# OpenReview forum: "AdvAgent: Controllable Blackbox Red-teaming on Web Agents"
_ICML.cc/2025/Conference — ICML 2025 poster_

### Official Review · Reviewer_oVuF · 2025-03-07

**Overall Recommendation:** 3

**Summary:**

The paper introduces AdvAgent, a black-box red-teaming framework that uses RL to optimize adversarial prompts, injecting them into HTML to mislead web agents. It achieves high attack success rates on GPT-4V and Gemini 1.5, revealing the limitations of prompt-based defenses.

**Claims And Evidence:**

Yes.

**Essential References Not Discussed:**

No.

**Ethical Review Concerns:**

The paper closely resembles preprint paper [1], and the name "AdvWeb" in Figure 5 from [1] has not been modified. Furthermore, the provided code differs from the publicly available code of [1]. If the author of [1] had submitted this paper, they should have used an anonymous code repository to upload the original code rather than significantly altering it. This raises concerns that the paper may be plagiarizing [1]. If it is confirmed that there was no intention to plagiarize, I will revise my evaluation accordingly.


[1] Xu, Chejian, et al. "Advweb: Controllable black-box attacks on vlm-powered web agents." arXiv preprint arXiv:2410.17401 (2024).

**Ethical Review Flag:**

Flag this paper for an ethics review.

**Ethics Expertise Needed:**

["Research Integrity Issues (e.g., plagiarism)"]

**Experimental Designs Or Analyses:**

Yes.

**Methods And Evaluation Criteria:**

Yes.

**Other Comments Or Suggestions:**

See weaknesses above.

**Other Strengths And Weaknesses:**

Strengths：

1. The paper proposes a novel black-box attack framework combining RL with adversarial prompt injection, which is relevant to web agent security.
2. Experimental results demonstrate high ASR across different web domains.

Weaknesses：

1. The paper lacks theoretical justification, as it does not adequately explain why DPO was chosen over other RL methods.
2. The evaluation scope is limited, with experiments focusing on specific tasks, making it unclear whether the method generalizes well.
3. There is no analysis of computational cost, leaving the efficiency of AdvAgent for real-world applications unaddressed.
4. The defense analysis is weak, as it only evaluates prompt-based defenses without exploring stronger security mechanisms like robust optimization or model distillation.
5. There is an issue in Figure 5, which illustrates the prompt optimized by AdvWeb rather than the proposed method, AdvAgent.

**Questions For Authors:**

See weaknesses above.

**Relation To Broader Scientific Literature:**

NA

**Theoretical Claims:**

Yes.

---

> ### Author Rebuttal · Authors · 2025-04-01
>
> We sincerely thank the reviewer for their thoughtful and constructive feedback. Below, we address each of the concerns raised:
>
> > Q1: More clarifications on the usage of DPO optimization in AdvAgent.
>
> Thank you for the interesting question! We'd like to clarify that our reinforcement learning setup only receives binary signals as rewards. Specifically, we assign a binary reward based on whether the agent successfully performs the targeted adversarial action (e.g., selecting a target stock). A successful misbehavior is labeled as positive; otherwise, it is negative. These binary labels make other RL optimization methods such as GRPO—which rely on continuous and dense process reward signals [2]—unstable. However, binary feedback is well-suited for constructing pairwise preference data. Therefore, we adopt DPO as a more signal-efficient approach for training our adversarial prompter model.
>
> > Q2: Expanded evaluation tasks.
>
> We appreciate this insightful suggestion and fully agree that broader evaluation is essential for demonstrating the generality of AdvAgent. While our primary evaluation focuses on SeeAct agent with the Mind2Web dataset—selected for its realism and established use in the web agent community—we have extended our study to include the BrowserGym agent [3] in the WebArena environment [4], which offers a distinct set of agent architecture and web environment.
>
> As shown in Table A below, AdvAgent achieves high attack success rates (ASR) when applied to BrowserGym, confirming that our attack strategy generalizes across different web agents and environments.
>
> Table A: Attack results on 3 domains in WebArena with BrowserGym agent
>
> | Agent       | D1   | D2   | D3   | Mean ± Std    |
> |-------------|------|------|------|----------------|
> | BrowserGym  | 95.1 | 80.8 | 44.6 | 73.5 ± 21.2    |
>
> These results reaffirm that AdvAgent generalizes across both agent designs (e.g., SeeAct, BrowserGym) and web interaction environments (e.g., Mind2Web, WebArena). We will include these expanded experiments and corresponding analysis in the final version of the paper to better highlight the flexibility and impact of our framework.
>
> > Q3: Clarifications of computational cost.
>
> Thank you for the suggestion. Our method is efficient—generating an adversarial prompt takes 2.2 seconds per task on average. Unlike optimization-based baselines requiring task-specific tuning, it uses a single forward pass, making it suitable for real-time red-teaming. We will include this in the final paper.
>
> > Q4: Evaluations of stronger defense strategies.
>
> We appreciate the reviewer’s suggestion to evaluate stronger defenses. To this end, we tested our AdvAgent framework on the SeeAct agent using GPT-4o-mini as the backend model. GPT-4o-mini is specifically optimized for robustness against jailbreak-style attacks through enhanced alignment techniques, including instruction hierarchies that guide the model to avoid unsafe or conflict behaviors during inference. These instruction hierarchies are designed to suppress potentially harmful actions by aligning the model’s response with different hierarchies of roles. As part of the model’s robust training, behaviors indicative of jailbreaking are explicitly suppressed.
>
> The results, summarized in Table B below, show that instruction hierarchies provide stronger defense compared to prompt-based strategies previously shown in our paper. Nonetheless, AdvAgent still achieves a relatively high average attack success rate (ASR) of ~59%, indicating that even models with advanced alignment mechanisms remain vulnerable to our attacks.
>
> Table B: Effectiveness of instruction hierarchy defense
>
> | SeeAct Backend | D1    | D2    | D3    | D4    | Mean ± Std    |
> |----------------|-------|-------|-------|-------|----------------|
> | GPT-4V         | 100.0 | 94.4  | 97.6  | 98.0  | 97.5 ± 2.0     |
> | GPT-4o-mini    | 56.9  | 60.5  | 83.3  | 34.1  | 58.7 ± 17.4    |
>
>
>
> [1] Shao, Zhihong, et al. "Deepseekmath: Pushing the limits of mathematical reasoning in open language models." arXiv preprint arXiv:2402.03300 (2024).
>
> [2] Lightman, Hunter, et al. "Let's verify step by step." ICLR 2023.
>
> [3] Drouin, Alexandre, et al. "Workarena: How capable are web agents at solving common knowledge work tasks?." ICML 2024.
>
> [4] Zhou, Shuyan, et al. "Webarena: A realistic web environment for building autonomous agents." ICLR 2024.

---

> > ### Comment · Reviewer_oVuF · 2025-04-04
> >
> > Thank you for your detailed response and clarifications. I appreciate the effort to address my concerns, particularly the expanded experiments and additional explanations. However, the paper would benefit from a more robust theoretical foundation to better support the findings, which would enhance its overall impact. As such, I am inclined to maintain my current evaluation. While the work shows promise, it would greatly benefit from further theoretical exploration in future research directions.

---

> > > ### Author Response · Authors · 2025-04-04
> > >
> > > We sincerely thank the reviewer for taking the time to read our response and for the thoughtful follow-up! We’re glad to hear that the expanded experiments and clarifications were helpful.
> > >
> > > We fully agree that developing a stronger theoretical foundation is a valuable and exciting direction for future work. However, the primary objective of this paper is to propose and empirically validate a practical red-teaming framework for web agents. To that end, we focus on demonstrating the effectiveness of AdvAgent through extensive evaluations. Our results show that AdvAgent consistently outperforms strong baselines across diverse attack scenarios, remains highly effective even against well-aligned models like GPT-4o-mini with instruction hierarchy, and generalizes well across different environments (e.g., WebArena) and agent architectures (e.g., BrowserGym), underscoring its robustness and real-world applicability. While theoretical analysis is beyond the scope of this work, we will explicitly include a discussion of it as a promising future direction in the revised version.
> > >
> > > Once again, we thank the reviewer for their thoughtful feedback and constructive suggestions.

---

### Official Review · Reviewer_WJQ8 · 2025-03-15

**Overall Recommendation:** 4

**Summary:**

This paper introduces AdvAgent, a black-box red-teaming framework designed to systematically uncover vulnerabilities in foundation model-based web agents, which are increasingly used to automate complex tasks but also pose significant security risks. The key contribution of AdvAgent is its use of a reinforcement learning-based pipeline to train an adversarial prompter model. This model generates optimized adversarial prompts that exploit weaknesses in web agents while maintaining stealth and controllability. The framework is evaluated extensively, demonstrating high success rates in compromising state-of-the-art GPT-4-based web agents across various tasks. The study also reveals that existing prompt-based defenses offer limited protection, leaving agents highly vulnerable to AdvAgent's attacks, with attack success rates (ASRs) exceeding 88.8% even after applying defenses.

**Claims And Evidence:**

This paper made various claims upon the constraints and challenges on the attacks against web agent. Some are supported by existing literature and others are based on the authors' own knowledge. Based on my background of knowledge, I think those claims generally make sense.

**Essential References Not Discussed:**

N/A

**Experimental Designs Or Analyses:**

The overall experimental designs or analyses are sound to me. However, I would suggest more comprehensive analysis on agent models and datasets to show the generality of the approach.

**Methods And Evaluation Criteria:**

This work leverages reinforcement learning from AI feedback (RLAIF)-based framework for black-box red-teaming against web agents. The overall pipeline is reasonable. The key concern to me is that this work only uses SeeAct as the victim model, and the experiments are performed on only one dataset, Mind2Web. The generality of the results is not clear. Given that fact that there are many choices of agent models and benchmark datasets, the paper could be strengthened with more comprehensive evaluation.

**Other Comments Or Suggestions:**

N/A

**Other Strengths And Weaknesses:**

N/A

**Questions For Authors:**

N/A

**Relation To Broader Scientific Literature:**

This paper is closely related to the broader research area of (M)LLM agents and safety. Existing studies have revealed various security risks of the agents. This paper could facilitate existing studies with a controllable black box red-teaming framework. The framework could be generally effective across related fields.

**Theoretical Claims:**

There is no theoretical claim in this paper.

---

> ### Author Rebuttal · Authors · 2025-04-01
>
> We thank the reviewer for their positive assessment and thoughtful feedback. We appreciate your recognition of the novelty and effectiveness of AdvAgent, and your acknowledgment that the proposed RL-based framework is reasonable and promising. Below, we respond to your main concern regarding generality:
>
> > Q1: The paper only uses SeeAct agent and the Mind2Web dataset. The generality of the results is not clear. More comprehensive evaluation across agents and datasets is suggested.
>
> We appreciate this insightful suggestion and fully agree that broader evaluation is essential for demonstrating the generality of AdvAgent. While our primary evaluation focuses on SeeAct agent with the Mind2Web dataset—selected for its realism and established use in the web agent community—we have extended our study to include the BrowserGym agent [1] in the WebArena environment [2], which offers a distinct set of agent architecture and web environment.
>
> As shown in Table A below, AdvAgent achieves high attack success rates (ASR) when applied to BrowserGym, confirming that our attack strategy generalizes across different web agents and environments.
>
> Table A: Attack results on 3 domains in WebArena with BrowserGym agent
>
> | Agent       | D1   | D2   | D3   | Mean ± Std    |
> |-------------|------|------|------|----------------|
> | BrowserGym  | 95.1 | 80.8 | 44.6 | 73.5 ± 21.2    |
>
> These results reaffirm that AdvAgent generalizes across both agent designs (e.g., SeeAct, BrowserGym) and web interaction environments (e.g., Mind2Web, WebArena). We will include these expanded experiments and corresponding analysis in the final version of the paper to better highlight the flexibility and impact of our framework.
>
> [1] Workarena: How capable are web agents at solving common knowledge work tasks? ICML 2024.
>
> [2] Webarena: A realistic web environment for building autonomous agents. ICLR 2024.

---

> > ### Comment · Reviewer_WJQ8 · 2025-04-06
> >
> > Thanks for the response. I appreciate the authors' efforts in addressing my major concern on the generality.
> >
> > After reading the other reviewers' comments and the author response, I think it is reasonable to perform HTML injection in this work as a widely used paradigm (though I agree that visual attack would be quite promising, too -- but this could be falling into different reseach lines due to the diversity of backbone LLMs or MLLMs).
> >
> > I believe that this is a solid paper.

---

### Official Review · Reviewer_NKrr · 2025-03-15

**Overall Recommendation:** 3

**Summary:**

The paper introduces AdvAgent, a black-box red-teaming framework designed to red team web agents against prompt injection attacks.These agents, while enhancing productivity, pose security risks due to their autonomous decision-making capabilities. The method first starts by collecting the dataset of successful/unsuccessful prompt injections by prompting target model. Afterwards, AdvAgent employs SFT followed by DPO to train an adversarial prompter model on that dataset. The framework is evaluated against GPT-4V based web agent (only text), achieving high success rates in various tasks, and revealing that these models are susceptible to these types of attacks. The study highlights the need for stronger security measures to protect web agents from adversarial attacks.

**Claims And Evidence:**

- Vulnerability of web agents and need for stronger defenses: The paper claims that current web agents, particularly those based on foundation models like GPT-4, are vulnerable to prompt injection attacks (or in general adversarial attacks), which can lead to severe consequences in high-stakes domains such as finance and healthcare. Prompt-based defense is certainly not enough. I agree with this claim and evidences provided in this paper partially confirm this. That said, community is developing various methods to fight against it. For example, instruction hierarchy proposed by OpenAI is specifically designed for this type of scenario and I strongly encourage authors to consider that defense mechanism in their experiments. It is implemented in GPT-4o-mini: https://openai.com/index/gpt-4o-mini-advancing-cost-efficient-intelligence/

- AdvAgent Framework: It introduces AdvAgent, a novel black-box red-teaming framework that effectively identifies and exploits vulnerabilities in web agents by generating adversarial prompts through a SFT+DPO pipeline. The position as a general method. However, I strongly disagree that this argument holds. Although web agents became a buzz word and there is no standard on the mode of its operations, web agents seem to be converging towards screenshot-based operation mode (OpenAI Operator, Claude computer use agent, etc.). And given the fact that the method can only operate on non-visual elements (via HTML injection), I don't think it can be considered as general as authors claim. Indeed, experiments do not contain any proper web agent, only scaffolding around GPT-4V. I'd suggest positioning this paper as red teaming method for text-based agents (e.g. tool usage or via API).

- High Success Rates: The paper claims that AdvAgent achieves high attack success rates against state-of-the-art web agents across diverse tasks, significantly outperforming existing baseline methods. I don't think that the comparison with GCG was conducted in a fair way. Why LLaVa was used as a proxy whitebox model? As I mentioned above, the considered threat model only applicable to text-based models. Ones needs to optimize GCG in this particular setting (using the same inputs and targets). I simply cannot believe that GCG would achieve 0 in this scenario! Moreover, since the agents are text-based, I'd also include some of the chatbot attack methods as baselines.

- Limited Defense Effectiveness: It asserts that current prompt-based defenses provide limited protection against the types of attacks generated by AdvAgent, indicating a critical need for more robust security measures. While I partially agree with this statement, more advanced defense mechanisms (e.g. instruction hierarchy as described above) necessary to validate this.

- Stealthiness and Controllability: AdvAgent is claimed to maintain stealthiness and controllability in its attacks, allowing adversaries to modify attack targets easily without re-optimization. While this is true for the method itself, threat model assumes Attacker can control arbitrary HTML elements of the web page. This is only achievable for untrusted web resources (e.g. attacker creates its own website). However, current web agents operates only on white-listed web apps, where attacker has a limited control.

**Essential References Not Discussed:**

N/A

**Experimental Designs Or Analyses:**

see methods and evaluation criteria; claim and evidences

**Methods And Evaluation Criteria:**

Overall, evaluation metrics are fine to me. I've listed several issues above and some of them applicable here (baselines, defense mechanisms). Moreover, I strongly believe that considering GPT-4 based agent is not enough as evidence. Experiments should also include "real" web navigation agents available at hand.

**Other Comments Or Suggestions:**

--------- Post-Rebuttal -----------

Authors provided additional experiments with Instruction Hierarchy (IH) defense mechanism. IH seems to significantly reduce the efficacy of the attack, although overall ASR is still high (~60%). That said, results on strong baselines such as GCG was not presented.

Moreover, I'm still not sure how realistic the threat model described above. Afaik, all commercial agents do not use invisible html elements as observations so it is unclear if the method will be relevant for that.

With this big two disclaimers, I'm increasing my score as the paper seems to be the first in this series and ML community might benefit on that. Though my concern on threat model is quite important.

**Other Strengths And Weaknesses:**

discussed in other sections above

**Questions For Authors:**

- How do you handle the case if there is a sparse signal for attacks? I'd imagine future agent will be more robust and the method receives more negative signals rather than positive ones. This is a fundamental problem in RL.

- I did not get why optimization search space is reduced? You still need to optimize over vocabulary of tokens (to generate prompt) + location to put the attack?

- what exactly are positive/negative rewards? Is it a binary indicator based on model output? Please, define it explicitly.

**Relation To Broader Scientific Literature:**

- the method is addressing an important problem and novel in the aspect of proposing red-teaming methodology for web agents.

**Theoretical Claims:**

N/A

---

> ### Author Rebuttal · Authors · 2025-04-01
>
> We thank the reviewer for their thoughtful and constructive feedback. We appreciate the recognition of the novelty and importance of our work in red-teaming web agents. We address the specific concerns and suggestions below:
>
> > Q1: Generality of AdvAgent and experiments on additional web agents.
>
> We appreciate the insightful comment! We acknowledge that purely screenshot-based web agents (e.g., OpenAI’s Operator) have recently demonstrated promising performance in everyday tasks like ticket booking. However, we would like to emphasize that text- and screenshot-based web agents remain one mainstream approach, particularly for tasks that demand precise data perception, such as data management in ServiceNow [1] or spreadsheet manipulation in WebArena [2]. These applications involve handling large volumes of structured data, where agents leveraging both textual/code and screenshots show superior performance [1,3,4]. Therefore, AdvAgent is broadly applicable to these generalist web agents, which are especially well-suited for data-intensive scenarios.
> Empirically, beyond evaluating SeeAct, we also test the BrowserGym agent [1] in the WebArena environment [2] to further validate the effectiveness of AdvAgent as a general red-teaming framework. Results and analysis are shown in Q1 of our response to Reviewer WJQ8, where we show AdvAgent is general and able to attack other web agents in other environments.
>
> > Q2: More clarifications on GCG and evaluation on additional baselines.
>
> Thank you for the valuable comment! Our GCG optimization uses the same inputs (HTML + SeeAct prompt [3]) and targets the same malicious behaviors as AdvAgent. We chose LLaVA as the proxy model since it's the default in SeeAct [3] and performs well in the context. However, its poor results in our setting highlight limited transferability to GPT-4V backends, pointing to the need for better black-box optimization methods.
>
> In addition to the two agent-specific black-box optimization methods (Agent-Attack and InjecAgent), we evaluate two additional state-of-the-art general-purpose black-box attack methods for LLMs: PAIR [5] and TAP [6]. We attack the SeeAct agent with GPT-4V as the backbone. As shown in Table A, the results indicate that AdvAgent still outperforms strong black-box attack methods for LLMs.
>
> Table A: Additional baselines
>
> |Algorithm|D1|D2|D3|D4|Mean±Std|
> |-|-|-|-|-|-|
> |PAIR|25.7|17.1|31.4|28.6|25.7 ± 5.3|
> |TAP|51.4|57.1|80.0|37.1|56.4 ± 15.4|
> |AdvAgent|100.0|94.4|97.6|98.0|97.5 ± 2.0|
>
> > Q3: Testing with defenses such as instruction hierarchy.
>
> Thank you for the suggestion. Due to space constraints, we address this question in Q4 of our response to Reviewer oVuF, where we show that AdvAgent still achieves a relatively high ASR against instruction hierarchy.
>
> > Q4: More clarifications on the threat model.
>
> We appreciate the reviewer’s concern. Like [7, 8], our threat model assumes agents operate in untrusted or semi-trusted environments—e.g., browsing third-party sites or interacting with embedded services—where supply-chain attacks and hidden HTML manipulations (e.g., via `aria-label` or `id`) are realistic risks. We acknowledge this may not apply to fully whitelisted settings and will clarify the scope in the revision.
>
> > Q5: Clarification on reward signal and sparse feedback.
>
> Thank you for these interesting questions!
>
> We assign a binary reward: positive if the agent performs the targeted adversarial action (e.g., selects the specified stock), negative otherwise.
>
> When positive signals are extremely sparse—especially with highly safety-aligned models—we train the adversarial prompter using online RL to dynamically collect signals during training. In parallel, we incorporate off-policy guidance [9, 10] from a less safety-aligned model and easier data points. This proxy supervision helps the model learn the structure of successful jailbreaks more efficiently. Online exploration further amplifies positive signals over time. We will expand on this in Section 4.2 in the revision.
>
> > Q6: Clarification on the optimization search space reduction
>
> Thank you for the question. We reduce the search space by fixing the injection position, avoiding the need to optimize over multiple locations.
>
> [1] Workarena: How capable are web agents at solving common knowledge work tasks? ICML 24.
>
> [2] Webarena: A realistic web environment for building autonomous agents. ICLR 24.
>
> [3] Gpt-4v(ision) is a generalist web agent, if grounded. ICML 24.
>
> [4] Agent workflow memory. arXiv 2024.
>
> [5] Jailbreaking black box large language models in twenty queries. arXiv 2023.
>
> [6] Tree of attacks: Jailbreaking black-box llms automatically. NeurIPS 2024.
>
> [7] Eia: Environmental injection attack on generalist web agents for privacy leakage. ICLR 25.
>
> [8] Attacking Vision-Language Computer Agents via Pop-ups. arXiv 2024.
>
> [9] Hindsight experience replay. NeurIPS 2017.
>
> [10] Reinforcement learning with sparse rewards using guidance from offline demonstration. ICLR 22.

---

> > ### Comment · Reviewer_NKrr · 2025-04-02
> >
> > Thanks for submitting rebuttal and performing additional experiments. I really appreciate that! My impression form the results on gpt-4o-mini is that IH works reducing ASR almost two times (although not clear since we don't have results without it). Also, what  are the performances of baselines (gcg, tap, etc.) on gpt-4o-mini?

---

> > > ### Author Response · Authors · 2025-04-04
> > >
> > > We thank the reviewer for carefully reading our rebuttal and for the thoughtful follow-up! We're glad the additional experiments were helpful, and we appreciate the suggestion to report baseline performance on GPT-4o-mini. Below, we provide further clarifications and new results.
> > >
> > > > Instruction Hierarchy Effectiveness and Baseline Performance on GPT-4o-mini
> > >
> > > Indeed, instruction hierarchy (IH) plays a significant role in reducing the attack success rate (ASR). In our main experiments with GPT-4V as the backend, AdvAgent achieves an ASR of 97.5%. When switching to GPT-4o-mini, which incorporates IH and other alignment mechanisms, the ASR drops to 58.7%. This demonstrates that IH can significantly reduce adversarial behaviors and provide meaningful mitigation. However, the fact that AdvAgent still succeeds in nearly 60% of cases under this stronger defense highlights its continued effectiveness and adaptability, even against models with advanced alignment mechanisms like instruction hierarchy.
> > >
> > > To further contextualize this, we additionally evaluated baseline methods on GPT-4o-mini under the same settings. As shown in Table A, AdvAgent consistently outperforms all baseline methods, including general-purpose black-box attacks (PAIR, TAP) and agent-specific attacks (Agent-Attack, InjecAgent). The performance gap is substantial—AdvAgent improves over the best baseline (InjecAgent) by 28% on average, demonstrating its robustness even against agents with stronger alignment defenses.
> > >
> > > Table A: ASR (%) on SeeAct agent powered by **GPT-4o-mini with instruction hierarchy defense**.
> > >
> > > | Algorithm     | D1    | D2    | D3    | D4    | Mean ± Std    |
> > > |---------------|-------|-------|-------|-------|----------------|
> > > | PAIR          | 2.9   | 0.0   | 2.9   | 0.0   | 1.5 ± 1.5      |
> > > | TAP           | 14.3  | 0.0   | 8.6   | 0.0   | 5.7 ± 6.1      |
> > > | Agent-Attack  | 17.14 | 28.57 | 57.14 | 8.57  | 27.9 ± 18.3    |
> > > | InjecAgent    | 28.57 | 33.33 | 38.10 | 22.86 | 30.7 ± 5.6     |
> > > | AdvAgent      | 56.9  | 60.5  | 83.3  | 34.1  | 58.7 ± 17.4    |
> > >
> > > For ease of comparison, in Table B, we provide the results previously presented in the paper and earlier in our rebuttal for the GPT-4V backend (without IH). This comparison highlights the impact of instruction hierarchy: all methods see a sharp ASR drop under GPT-4o-mini, but AdvAgent remains significantly more effective than all alternatives across both settings.
> > >
> > > Table B: ASR (%) on SeeAct agent powered by **GPT-4V without instruction hierarchy defense**.
> > >
> > > | Algorithm     | D1    | D2    | D3    | D4    | Mean ± Std    |
> > > |---------------|-------|-------|-------|-------|----------------|
> > > | PAIR          | 25.7  | 17.1  | 31.4  | 28.6  | 25.7 ± 5.3     |
> > > | TAP           | 51.4  | 57.1  | 80.0  | 37.1  | 56.4 ± 15.4    |
> > > | Agent-Attack  | 26.4  | 36.0  | 61.2  | 58.0  | 45.4 ± 14.6    |
> > > | InjecAgent    | 49.6  | 47.2  | 73.2  | 87.2  | 64.3 ± 16.7    |
> > > | AdvAgent      | 100.0 | 94.4  | 97.6  | 98.0  | 97.5 ± 2.0     |
> > >
> > > We sincerely thank the reviewer again for encouraging us to strengthen both the experimental scope and comparative clarity. We hope this clarifies the performance impact of instruction hierarchy and further reinforces the strength and adaptability of AdvAgent, even in modern, defense-aware deployment settings.

---

### Decision · Program_Chairs · 2025-05-01

**Decision:**

Accept (poster)

**Comment:**

This paper introduces AdvAgent, a new black-box red-teaming framework for web agents. AdvAgent first collects a dataset of successful/unsuccessful prompt injections, and uses this dataset to train the adversarial prompter via SFT/DPO. The authors evaluate AdvAgent against GPT-4-based agents, and finds that existing prompting-based defenses do not offer adequate protection.

Reviewers generally agreed that the paper introduced a novel and effective automated black-box attack against web agents, but also raised several weaknesses:
1. The attack relies on injecting invisible elements into the webpage's HTML, which is not applicable to screenshot-based agents such as OpenAI's operator and Claude Computer Use Agent. This severely limits the method's generality.
2. Lack of comparison to other automated attacks and stronger defenses. Authors included evaluation of PAIR, TAP, InjecAgent, etc., as well as the instruction hierarchy defense in the rebuttal.
3. Lack of evaluation in other web agent settings. Authors included evaluation of the BrowserGym agent in WebArena in the rebuttal.

Overall, AC finds that the weaknesses have been sufficiently addressed by the rebuttal experiments, and the paper is ready for publication at ICML.